# Tracking cropland transitions: A comparative analysis of U.S. land cover change data

**Gray Martin**[ORCID][1]*, **Kemen Austin**[2], **Tyler Lark**[ORCID][3], **Stanley Lee**[1], **Christopher M. Clark**[4]

**1** RTI International, Research Triangle Park, North Carolina, United States of America, **2** Global Conservation Program, Wildlife Conservation Society, Bronx, New York, United States of America, **3** Nelson Institute Center for Sustainability and the Global Environment (SAGE) and Great Lakes Bioenergy Research Center, University of Wisconsin-Madison, Madison, Wisconsin, United States of America, **4** Center for Public Health and Environmental Assessment, Office of Research and Development, U.S. Environmental Protection Agency, Washington, DC, United States of America

* gdmartin@rti.org

## Abstract

There are a growing number of land cover data available for the conterminous United States, supporting various applications ranging from biofuel regulatory decisions to habitat conservation assessments. These datasets vary in their source information, frequency of data collection and reporting, land class definitions, categorical detail, and spatial scale and time intervals of representation. These differences limit direct comparison, contribute to disagreements among studies, confuse stakeholders, and hamper our ability to confidently report key land cover trends in the U.S. Here we assess changes in cropland derived from the Land Change Monitoring, Assessment, and Projection (LCMAP) dataset from the U.S. Geological Survey and compare them with analyses of three established land cover datasets across the coterminous U.S. from 2008-2017: (1) the National Resources Inventory (NRI), (2) a dataset Lark *et al*. 2020 derived from the Cropland Data Layer (CDL), and (3) a dataset from Potapov *et al*. 2022. LCMAP reports more stable cropland and less stable noncropland in all comparisons, likely due to its more expansive definition of cropland which includes managed grasslands (pasture and hay). Despite these differences, net cropland expansion from all four datasets was comparable (5.18-6.33 million acres), although the geographic extent and type of conversion differed. LCMAP projected the largest cropland expansion in the southern Great Plains, whereas other datasets projected the largest expansion in the northwestern and central Midwest. Most of the pixel-level disagreements (86%) between LCMAP and Lark *et al*. 2020 were due to definitional differences among datasets, whereas the remainder (14%) were from a variety of causes. Cropland expansion in the LCMAP likely reflects conversions of more natural areas, whereas cropland expansion in other data sources also captures conversion of managed pasture to cropland. The particular research question considered (e.g., habitat versus soil carbon) should influence which data source is more appropriate.

## Introduction

Agriculture is a dominant driver of global environmental change, with substantial documented impacts on land cover and ecosystem processes [1,2]. Monitoring the ways in which

**Data availability statement:** All data and calculations are available from the following repository: Data and calculations associated with "Tracking cropland transitions: a comparative analysis of U.S. land cover change data" (https://zenodo.org/records/13909562). The LCMAP dataset and associated reference materials are available from https://www.usgs.gov/special-topics/lcmap/lcmap-data-access. The data associated with Lark et al. 2020 are available from https://zenodo.org/records/3905243#.YpZSQ6jMKUk. The data associated with Potapov et al. 2021 are available from https://glad.umd.edu/dataset/croplands.

**Funding:** RTI received financial support for this work from EPA's Office of Research and Development (https://www.epa.gov/aboutepa/about-office-research-and-development-ord) under contract 68HERD20A0004/68HERH22F0010. The views presented are those of the authors and do not necessarily represent the views or policies of the US Environmental Protection Agency. The funders participated in the study design, interpretation of results, and in manuscript preparation.

**Competing interests:** The authors have declared that no competing interests exist.

agricultural landscapes change over time is critical for understanding the causes and estimating the consequences of observed land transformations and managing the net impacts of land conversion.

In the U.S., as in the rest of the world, demands for food, fiber, feed, and fuel put pressure on the land area dedicated to crop production [1,3]. It is vital to understand how these pressures influence land use across scales. To support this effort, U.S. government agencies have developed several national scale programs and spatially explicit datasets tracking agricultural land cover and land use. Datasets in this category include the USDA National Resource Inventory (NRI) [4], the USGS National Land Cover Database (NLCD) [5], the USDA Cropland Data Layer (CDL) [6], and, more recently, the USGS Land Change Monitoring, Assessment and Projection (LCMAP) [7] data product. Related datasets also include the National Agricultural Imagery Program (NAIP) [8] and the National Agriculture Statistics Service's (NASS) annual surveys and Census of Agriculture [9], both developed by the U.S. Department of Agriculture (USDA).

These datasets vary in their source data (e.g., earth observation imagery, field sampling, aerial photo interpretation, farmer surveys), spatial scale of representation (e.g., 30-m, county), frequency of data collection and reporting (e.g., annual, 5-year epoch), and land class definitions and categorical detail, among other characteristics. As a result, there are reported discrepancies among products. These discrepancies lead to confusion in the academic community and among policy makers – notably, in terms of how much agricultural expansion is actually occurring in the U.S. and where those changes are occurring.

Despite the importance of land cover data in the US across a number of critical applications, there have been very few studies directly comparing estimates among datasets or investigating the character and underlying reasons for these discrepancies. Several studies have examined the robustness of—and best practices for using—the CDL data nationally [10,11], for the 20 states for which CDL was available in 2007 [12], and in specific regions of the U.S. [13,14]. However, the more recently developed LCMAP has not yet benefited from the same level of assessment.

This study focuses on the LCMAP dataset (Collection 1.1), which is based on Landsat imagery, has a 30-meter spatial resolution, and is available annually from 1985 to 2019 [15]. LCMAP is one of the only national datasets with consistent representation annually going back to the 1980s, making it attractive for applications that require a longer historical time series. National coverage for the CDL does not begin until 2008, but large increases in corn acreage took place from 2006-2008 [16,17]. Furthermore, much of the research in the scientific literature examines biofuels as a driver of land use [3,14,18]; however, by 2008 roughly half of the increase in corn ethanol consumption in the U.S. had already occurred [19]. Thus, examining earlier years is critical to understand the effects of biofuels policy in the United States.

The goals of this study are twofold: first, to understand how estimates of cropland change differ across four different data products (LCMAP, NRI, Lark et al. 2020 (based on the CDL), and Potapov et al. 2022), and second, to examine which broad factors drive these differences. There are other datasets we could have included, but each of these datasets are included or omitted for specific reasons described below. To meet these goals, we harmonize the spatial and temporal aspects of the datasets to a common basis, using the LCMAP as the main point of comparison. We do not harmonize the class definitions because not only would that require re-deriving the original datasets, but also because these differences are the focus of this study—we are interested in how definitional differences lead to different dynamics and patterns of estimated cropland expansion. This will enable future users of these datasets to understand their strengths and limitations and how they compare to alternatives to inform

future research and analysis. This insight is critical, as these datasets are increasingly used to inform policies affecting land resources such as biofuel mandates, habitat conservation, and plans for climate change mitigation and adaptation.

## Methods

### Dataset selection and key attributes

In this study, we compare LCMAP-derived land change estimates with three other datasets: NRI, Lark *et al.* 2020, and Potapov et al. 2022. Since the primary goal of this study is an assessment of LCMAP, we do not prioritize comparisons among the other three datasets.

We chose NRI for its ubiquity and longevity as a data source for land use and land use change studies. NRI is the only sample-based dataset in our comparison, and thus it represents the closest estimate of actual cropland changes that does not rely on remote sensing. Because of this, some U.S. agencies prefer it for estimates of trends of agricultural expansion [17,19]. Though the data is spatially aggregated to the cropland reporting district (CRD) level, NRI provides high-quality land use change at 5-year intervals suitable for long-term analysis of land use change trends. Comparison with LCMAP data is hindered by the spatial aggregation, temporal availability, and definitional differences, which are discussed below.

We chose Lark *et al.* 2020 both because it is based on a commonly used foundational dataset—the CDL—and because it processed the CDL to remove irregularities in the raw output and simplified classes to optimize estimates of cropland change [3,10]. Additionally, this study has received much attention in the literature. Lark *et al.* 2020 also provides the methodological framework for calculating land use change from CDL land cover maps that this study applies to the LCMAP. Given the similarities in the spatial resolution, extent, and temporal availability between the underlying datasets—and the common pre-processing steps we applied (discussed below)—there are fewer barriers to direct comparison between Lark *et al.* and LCMAP, allowing us to highlight the critically important differences in definitions. To distinguish between the original publication and the dataset modified for use in this study, we will refer to the paper as Lark *et al.* 2020 and the derived land change estimates as Lark *et al.*

We chose Potapov *et al.* 2022 for its recency, global scope, and high spatial resolution (also 30 m). While Potapov *et al.* 2022 is available at the same resolution as LCMAP and Lark *et al.*, the data is aggregated to 4-year periods (similar to the NRI) of cropland extent that require alternative methods to compare land use change. To distinguish between the original publication and the dataset modified for use in this study, we will refer to the paper as Potapov *et al.* 2022 and the derived land change estimates as Potapov *et al.*

There are certainly other datasets that could have also been included but were not (e.g., National Agricultural Statistics Service (NASS), National Land Cover Database (NLCD), National Agriculture Imagery Program (NAIP)). In addition to study feasibility constraints, these were not used for specific reasons. We did not use NASS because it does not report total cropland and individual crops (which are not reported for all counties even if grown in that county) cannot be simply summed up due to the presence of double cropping and other data idiosyncrasies. We omit the NLCD because the CDL leverages NLCD for its non-crop classification and validation such that it would not be an independent comparison. The NAIP is not included because it is a repository of photographic images that have to be viewed one at a time (usually for validation purposes) and have not been stitched into a national or regional database to date to our knowledge.

Key attributes of the four datasets we used in this study are summarized in Table 1. In the methods, we detail our efforts to harmonize and indicate points of friction where harmonization was not an option without re-deriving the original datasets.

**Table 1. Overview of key attributes for land cover change data used to estimate cropland change in this study.**

| Dataset used to estimate cropland change | Land Change Monitoring, Assessment and Projection (LCMAP) | National Resource Inventory (NRI) | Cropland Data Layer (CDL, used in Lark *et al.* 2020) | Global Cropland Expansion in the 21st Century (published in Potapov *et al.* 2022) |
|---|---|---|---|---|
| Source | Landsat | Plot samples | Landsat, NLCD, and others | MODIS NPP and Landsat ARD |
| Resolution | 30 meter square pixels | Point-based (Aggregated to CRD in this application) | 30 meter square pixels | 30 meter square pixels |
| Thematic Resolution | 8 land cover classes | 6 land cover classes | 134 land cover classes | 2 land cover classes |
| Extent | CONUS | CONUS | CONUS | Global |
| Temporal Availability | Annual (1985-2019) | 5-year intervals (1982, 1997, 2002, 2007, 2012, 2017) | Annual (2008-2020) | 4-year epochs (2000-2003, 2004-2007, 2008-2011, 2012-2015, 2015-2019) |
| Dataset type | Land cover raster | Broad land cover classes by CRD | Crop-specific land cover raster | Cropland extent raster |

## LCMAP description

LCMAP is an annual land cover and change data product for the CONUS based on time series data from the Landsat record – specifically, data collected since 1985 with the 30-meter sensors of Thematic Mapper, Enhanced Thematic Mapper Plus, and Operational Land Imager [7]. This dataset enables the generation of continuous land cover trend information by characterizing surface conditions and land change using the Continuous Change Detection and Classification (CCDC) algorithm, which estimates the date at which a spectral time series significantly diverges from past patterns [20]. Although LCMAP has only been available since 2020, this dataset has found use across a range of applications [20–22].

The value of the LCMAP data product, unique from other datasets, stems from the CCDC algorithm's purpose to detect land cover change and avoid noise. Annual pixel transitions taken from LCMAP should better reflect actual land cover change on the ground [20]. For example, the spectral characteristics of a particular land cover type (grassland, for instance) may vary widely based on aridity, soil properties, or other modifying factors. LCMAP, through the CCDC algorithm, assesses whether the change in spectral properties is significant enough to constitute a change in land cover rather than a change in spectral characteristics within the same land cover type.

LCMAP classifies land cover into eight thematic categories (S2 Table) for its primary land cover classification. In addition to primary land cover class, LCMAP also includes an annual land cover change map and confidence values for all land cover classifications. For the purposes of this project, we use only the primary land cover class map as an input.

## Data processing and harmonization

**LCMAP data preprocessing.** LCMAP does not provide outputs that characterize the long-term behaviors of pixels over more than two years, necessitating a transition detection algorithm to classify pixels into relevant categories representing cropland expansion, cropland abandonment, intermittent cropland, and stable classes (stable cropland and stable noncropland). To construct this algorithm, we applied similar methods used in Lark *et al.* 2020 and followed best practices from Lark *et al.* 2017.

As described in Lark *et al.* 2020, we began by combining all LCMAP land cover classes other than cropland into a single non-cropland class, yielding a binary cropland extent map for each year (cropland or noncropland). We focus on 2008-2017 because all of the temporal comparisons with other datasets were done for that time period. We then combined these maps into a single trajectory layer with each pixel representing a temporal pattern of cropland status from 2008-2017. Since each pixel takes one of two values for each of the 10 years of

LCMAP data in this study, there are 1024 ($2^{10}$) values that each pixel in the temporal trajectory map can take, corresponding to 1024 unique 10-year temporal trajectories.

Next, we mapped each trajectory to one of five long-term pattern classes (LTPC) following the definitions established in Lark *et al.* 2020. Fig 1 below demonstrates examples of each LTPC. For the cropland expansion and cropland abandonment classes, we additionally determined the conversion year.

After classifying the temporal trajectories into one of the five long-term pattern classes, we applied a minimum mapping unit (MMU) to replace patches of LTPC smaller than 5 acres with the LTPC of their nearest neighbors. We accomplished this using methods matching those in the Lark *et al.* 2020 data processing code.

This pre-processing (derivation of LTPC and MMU) had already been performed on Lark *et al.*, so no additional pre-processing was needed for that dataset. For the NRI and Potapov *et al.*, these steps were not necessary or appropriate. For the NRI, the data are observations every five years and statistically scaled up to the CRD, so such a step is unwarranted (we assume cropland change reported in the NRI are not intermittent or temporary; there is no need for MMU because the estimates are point estimates statistically scaled up to the CRD). For Potapov *et al.*, these steps were also not necessary because the data were already reported for 4-year periods and the developers had already performed similar functions although through different analytical means (removal of cropland patches smaller than 0.5 hectares, as well as the use of decision tree ensembles to generate robust estimates of cropland) [21].

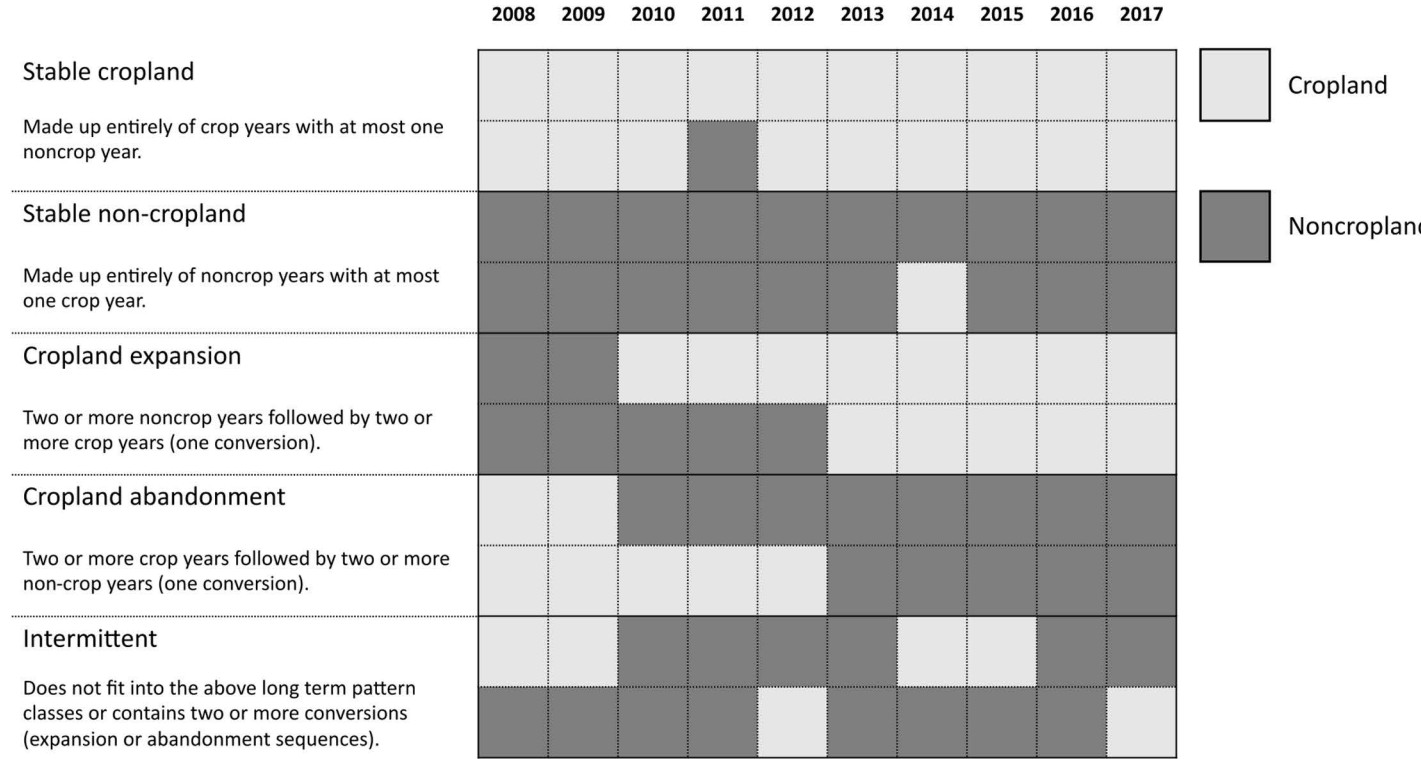

**Fig 1. Long-term pattern class (LTPC) definitions and examples of assignment of temporal trajectories to LTPCs.** These definitions are aligned with broad Land Use Change (LUC) classifications described in Lark et al. 2020. The second-to-last row indicates a scenario where a pixel transitioned from cropland to non-cropland in 2010 and then transitioned back to cropland in 2014. Because this pixel experienced more than one conversion over the study period, it would be assigned to the intermittent class. The last row indicates a scenario where a pixel remains non-cropland for the majority of the study period aside from two years. Because this pixel had more than a single year of cropland coverage, it would be assigned to the intermittent class instead of the stable non-cropland class.

**Definitional differences.**  One of the more challenging aspects of comparing among land cover datasets are discrepancies among underlying definitions of land cover classes [22,23]. One of the key differences among datasets is how land covered in grass and lightly managed is classified (e.g., as pasture, hay, or rangeland), whether in cropland or non-cropland (Table 2). All datasets include row crops in the definition of cropland and all datasets include natural grasslands and rangeland in the grassland category. Between these extremes, however, there are several land use and land cover types that these datasets categorize differently (Table 2). LCMAP defines cropland to include intensively managed pasture and hay but exclude unmanaged pastures, grassland, and rangeland (S1 Table). While the Potapov *et al.* definition of cropland similarly includes forage and hay, it excludes permanent managed pasture and perennial woody crops. The Lark *et al.* definition of cropland (derived from the CDL) excludes all hay and managed pasture lands in its definition of cropland. Finally, the NRI defines both cultivated and non-cultivated cropland as separate subclasses of cropland; the former category includes row crops and pastureland grown in rotation with row crops, while the latter category includes permanent machine-harvested hayland and horticultural cropland. The NRI further defines a category of managed pastureland which does not overlap the permanent hayland included in the uncultivated cropland class [24]. See Table 2 and S1 Table for more information.

It is not the purpose of this effort to re-derive the source data evaluated here. Rather, we aim to align these datasets as close as possible, spatially and temporally, and use their classifications to derive insights into land use change in the United States. It is the differences between the estimates from these sources that may yield insights on these various land use and land cover types.

**Aligning temporal differences with LCMAP.**  Another factor to consider when comparing land cover and land cover change maps are the dates of representation and the time intervals over which each dataset reports change. The period that most closely aligned for comparisons across datasets was 2010-2017 (7 years). Because cropland expansion is actually a two-or-more year phenomenon (depending on how many years a pixel is required to be non-crop prior to conversion), we required two-years of non-cropland prior to conversion to count as cropland expansion. This means that the first year of data used in the LCMAP was 2008 for a reported conversion in 2010. Note that Lark *et al.* 2020 also required two years of non-cropland prior to conversion with an exception for the first year (2008). Thus, although cropland expansion was reported in 2009 in Lark *et al.* 2020, we do not count this year in our comparisons to avoid an extra year in the Lark *et al.* estimates relative to LCMAP.

Because the data from the NRI and Potapov *et al.* are not available annually (Fig 2), we had to process these differently. For these datasets, we compare the average change in cropland

**Table 2. Cropland class definitions across datasets.**

| | LCMAP | NRI | Lark et al. | Potapov et al. |
|---|---|---|---|---|
| Cultivated land for the production of crops | Included | Included | Included | Included |
| Uncultivated or fallow land for the production of crops | Included | Included | Included | Included |
| Perennial woody crops including orchards and vineyards | Included | Included | Included | Excluded |
| Managed/permanent hay | Included | Included | Excluded | Included |
| Managed/permanent pasture | Included | Excluded | Excluded | Excluded |
| Unmanaged pasture or hay | Included | Excluded | Excluded | Excluded |
| Grassland, shrubland, and grazing land | Excluded | Excluded | Excluded | Excluded |
| CRP land | Partial | Excluded | Partial | Partial |

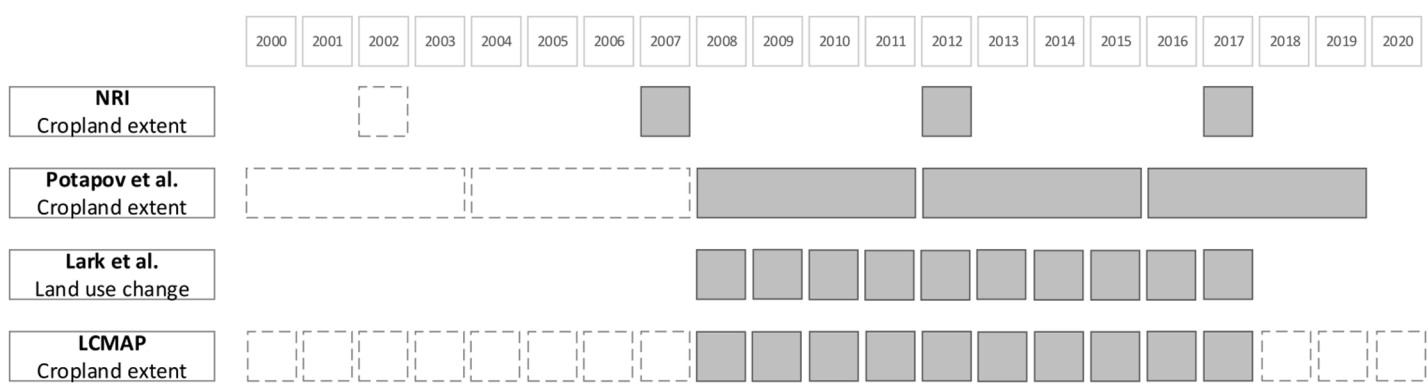

**Fig 2. Available cropland extent maps by product starting in 2000.** Intervals used in this comparison are shaded in darker colors. The NRI had an additional partial release in 2015 that is not included here. Additionally, cropland mapping for Potapov et al. was performed in four-year intervals.

acreage between the start and end years with the annual change values from LCMAP and Lark *et al.* 2020.

To account for different years of availability among the datasets in our comparison, we selected years from each dataset to align as closely as possible with the 2008-2017 interval. From NRI, we first used the 2007 and 2012 releases to estimate cropland expansion and abandonment within each CRD. Then we calculated an annual average for that 5-year period for each CRD and applied that to each year from 2007-2012. We took the same steps with the 2012 and 2017 releases to estimate land use change annually for 2012-2017. For the comparison with LCMAP and Lark *et al.* over 2010-2017, we only used the annual average values for the years in that interval. For Potapov *et al.* 2022, we processed these similarly to the NRI. First, we overlayed the cropland extent maps for the three primary mapping periods (2008-2011, 2012-2015, and 2016-2019) and calculated the number of pixels that converted to or from cropland to estimate cropland expansion, cropland abandonment, stable cropland, and stable noncropland, between each 4-year period. We then converted these to annual averages between adjacent 4-year periods. Then, for comparisons with LCMAP and Lark *et al.* for 2010-2017, we only used the annual average values for the years in that interval.

**Aligning spatial differences with LCMAP.** For comparisons between LCMAP, Lark *et al.*, and Potapov *et al.*, there were no additional alignments needed because they already at 30-m and pre-processed similarly. For comparisons with the NRI, data for each of the other three were summed to the CRD to align with the finest resolution of the NRI. The CRD spatial resolution is used for displaying purposes in the results to align all four datasets.

## Comparison metrics

Because LCMAP and Lark *et al.* are aligned spatially and temporally, we conducted additional comparisons between these datasets. We assessed the agreement between LTPCs at the level of the individual pixel and used this to determine the Kappa coefficient ($\kappa$), quantity disagreement, and allocation disagreement between the datasets. These metrics help pinpoint the underlying differences in the datasets that contribute to overall disagreement. Quantity disagreement is the difference in the proportions of categories between each dataset. Allocation disagreement is the difference in spatial allocation of classes, or pixel placement, between datasets. Quantity and allocation disagreement are mutually exclusive and sum to the overall disagreement, which is the percent of pixels in disagreement across all classes [25]. See Table 4 below for additional details.

**Table 3. Overview of temporal and definitional harmonization with LCMAP.**

| Dataset | Cropland definition harmonization | Time steps used for comparison | Spatial harmonization |
|---|---|---|---|
| LCMAP | Uses base LCMAP cropland class; see methods for LTPC definitions | Annual data from 2008-2017, processed for 2010-2017 comparisons | 30-m pixel data aggregated to CRD for display purposes. |
| NRI | Uses NRI Cultivated and Non-cultivated cropland class (which excludes CRP land and pastureland)<br>LTPC equivalences:<br>• Stable cropland and stable noncropland – quantity of land cover class remaining the same between surveys<br>• Cropland expansion – quantity of land cover moving from noncropland to cropland class between surveys<br>Cropland abandonment – quantity of land cover moving from cropland to noncropland between surveys | CRD-level land cover totals from three survey intervals (2007, 2012, and 2017)<br>Annualized for 2007-2017 and truncated to 2010-2017 for comparisons | None (already at CRD level). |
| Lark et al. | Uses definitions of LTPCs from Lark et al. 2020 | Annual data from 2008-2017, processed for 2010-2017 comparisons | 30-m pixel data aggregated to CRD for display purposes. |
| Potapov et al. | Uses base Potapov et al. 2022 cropland class<br>LTPC equivalences:<br>• Stable cropland and stable noncropland – no pixel change between epochs<br>• Cropland expansion – pixel moves from noncropland to cropland between epochs<br>Cropland abandonment – pixel moves from cropland to noncropland between periods | Cropland extent from three periods (2008-2011, 2012-2015, and 2016-2019)<br>Annualized for 2008-2019 and truncated to 2010-2017 for comparisons | 30-m pixel data aggregated to CRD for display purposes. |

**Table 4. Agreement and disagreement metrics for pixelwise comparison.**

| Metric | Definition | Notes |
|---|---|---|
| **Kappa coefficient** | Inter-rater reliability statistic that takes the possibility of chance agreement into account. | This metric measures agreement, while the other metrics measure disagreement |
| **Overall disagreement** | Fraction of pixels in disagreement across all classes. | |
| **Quantity disagreement** | Difference in the reported proportions of each class. Indicates disagreement on underlying assumptions of what pixels belong to which class. | Subset of overall disagreement (mutually exclusive with allocation disagreement) |
| **Allocation disagreement** | Difference in the spatial placement of pixels. Indicates disagreement on how classes are spatially distributed and how pixels are classified. | Subset of overall disagreement (mutually exclusive with quantity disagreement) |

## Results

### Aggregate cropland change 2008-2017

We found differences in the quantities of stable cropland and stable noncropland among the four datasets (Fig 3a). LCMAP reports higher values for stable cropland and lower values for stable noncropland compared with other sources. For Lark *et al.*, we found the opposite pattern, with the lowest stable cropland and highest stable noncropland. Values from NRI and Potapov *et al.* were intermediate and similar to one another.

With regards to the dynamic classes (cropland expansion and cropland abandonment), we see good agreement at CONUS level between LCMAP and Lark *et al.* (Fig 3b). However, NRI and Potapov *et al.* both report much higher quantities (2-4 times higher) of cropland abandonment and expansion. For net cropland expansion, however, all four estimates were relatively close to one another, estimating between 5.18 and 6.33 million acres of cropland expansion for the period between 2010 and 2017 (0.65-0.79 million acres per year, Table 5).

### Cropland expansion and abandonment over time

Over the 2010-2017 time interval (Fig 4), LCMAP reports the highest rates of cropland expansion and cropland abandonment in 2011 and in the latter years (2014 and 2015). This stands

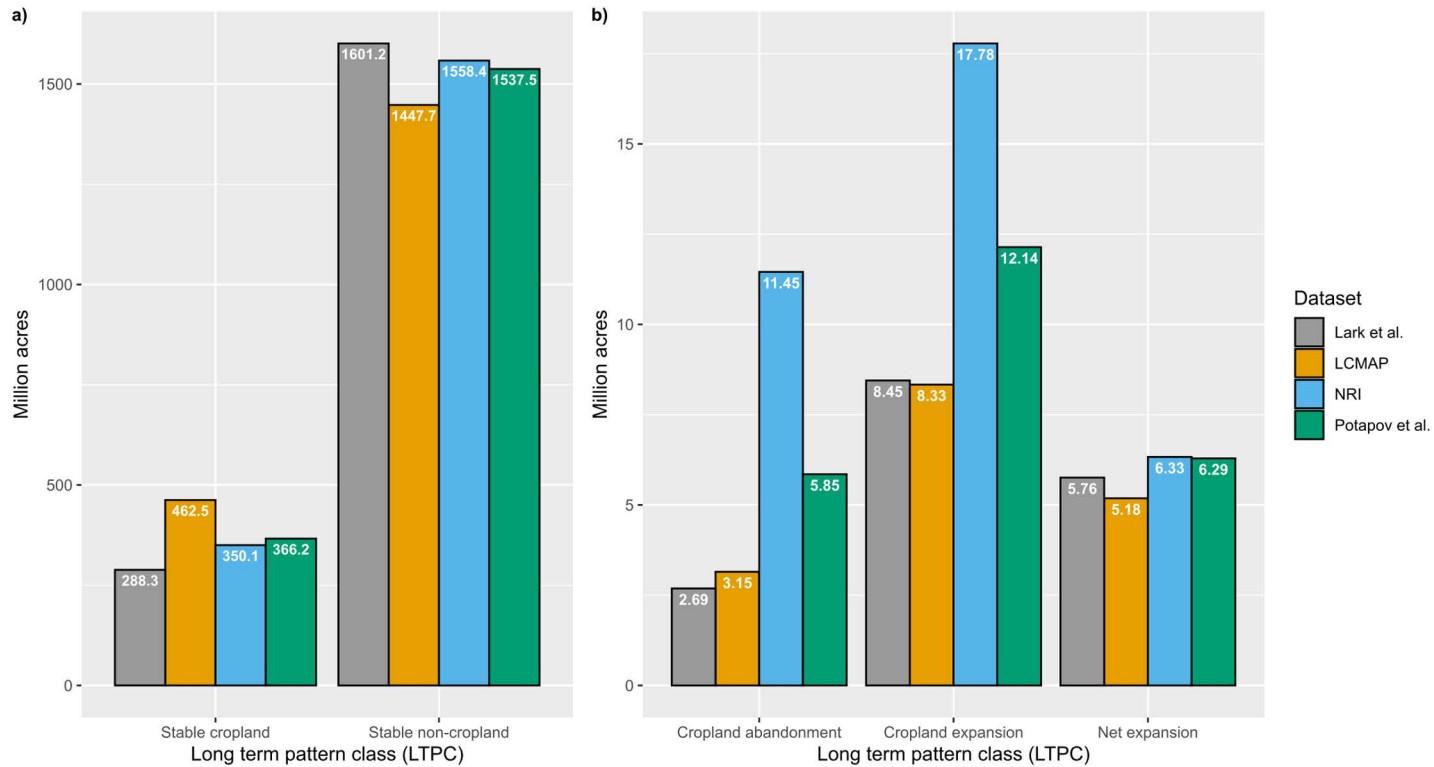

**Fig 3. High-level comparison of total change in stable classes (a) and dynamic classes (b) in thousands of acres from 2010-2017 at the CONUS level.**

in contrast to Lark *et al.*, which shows a decline in both cropland expansion and abandonment over time except in 2010 and 2011 (Fig 4). Similarly, the average annual expansion and abandonment rates derived from the NRI also declined. Potapov *et al.* shows an increase in expansion and abandonment in 2015. For both NRI and Potapov *et al.*, however, quantities are average annual rates derived from the total change across long intervals and thus cannot capture annual variance.

For net expansion, the patterns were dissimilar, with a generally decreasing trend in Lark *et al.* (aside from 2010-2011), an increasing trend in the LCMAP (aside from 2014-2017), an increasing trend for NRI, and little change for Potapov *et al.* These apparently contrasting trends make sense once the definitional differences are accounted for (see Discussion).

## Spatial patterns of cropland change

For cropland abandonment, we found relatively low levels (less than 1%) at the CRD level for LCMAP and Lark *et al.*, with higher levels scattered throughout the CONUS for NRI and Potapov *et al.* (Fig 5a–5d). For cropland expansion (Fig 5e–5h), we found higher rates than abandonment in the LCMAP and Lark *et al.* (greater than 3-4%), but these were located in different areas. Much of the cropland expansion in the LCMAP was in the southern Midwest (Fig 5e) while in Lark *et al.* it was more in the central and northern Midwest (Fig 5f). Rates of cropland expansion were higher for the NRI and Potapov *et al.* (2-6%) and more spatially aligned with Lark *et al.* Stable LTPCs were similar overall, with more cropland in the Midwest (Fig 5i–5l) and more noncropland in the West and East (Fig 5m–5p).

Spatial distributions of disagreement with LCMAP are similar between stable cropland and stable noncropland. In both comparisons, we see the most pronounced disagreement in

**Table 5. Acreage of each long-term pattern class in thousands of acres and percent difference from LCMAP reported values (in parentheses).**

| | LCMAP | NRI | Lark et al. | Potapov et al. |
|---|---|---|---|---|
| Cropland abandonment (1,000 acres) | 3,149 | 11,454 (264%) | 2,688 (-15%) | 5,851 (86%) |
| Cropland expansion (1,000 acres) | 8,333 | 17,783 (113%) | 8,447 (1%) | 12,141 (46%) |
| *Net expansion (1000 acres)* | 5,184 | 6,329 (22%) | 5,759 (11%) | 6,290 (21%) |
| Intermittent (1,000 acres) | 5 | NA | 17,953 (> 500%) | NA |
| Stable cropland (1,000 acres) | 462,450 | 350,051 (-24%) | 288,319 (-38%) | 366,207 (-21%) |
| Stable non-cropland (1,000 acres) | 1,447,736 | 1,558,427 (8%) | 1,601,232 (11%) | 1,537,461 (6%) |
| Total (1,000 acres) | **1,921,673** | **1,937,716** | **1,918,639** | **1,921,660** |

Note that because the NRI and Potapov *et al.* are not annual, there are no areas classified as intermittent.

Missouri, Iowa, and Kansas (centered roughly around Kansas City), Kentucky, and the eastern edge of Texas. We also observe widespread disagreement in North Dakota, South Dakota, and Wisconsin. We observe low disagreement along the East coast with one exception: in the comparison between reported stable non-cropland values between LCMAP and NRI, we observe hot spots of disagreement in Delaware, Maryland, and the coasts of Virginia and North Carolina. The regions with the most pronounced disagreement corresponds to areas where tallgrass, mixed, and shortgrass prairie are prevalent [26].

Difference maps with the LCMAP (Fig 6) highlight the differences among the datasets. For cropland abandonment, compared with LCMAP we observe similar amounts in Lark *et al.* (Fig 6a) and much higher amounts in NRI and Potapov *et al.* in the Dakotas, Kansas, Missouri, Indiana, Oklahoma, and Texas (Fig 6b and 6c). For cropland expansion, we observe disagreement between LCMAP and all three datasets in North and South Dakota, Iowa (near Des Moines), Missouri, Texas, and Oklahoma. LCMAP reports more cropland expansion in Texas and Oklahoma, while the other datasets report more cropland expansion in Missouri, Iowa, the Dakotas, Montana, and Kentucky. As noted earlier, there was more stable cropland and less stable noncropland in the LCMAP compared with other data sources (Fig 6g–6l).

## Disagreement measures

For the supplemental comparisons between LCMAP and Lark *et al.*, overall disagreement was low, with only 10.6% of pixels disagreeing; in other words, 89.4% of pixels agreed in assigned long-term pattern class between the two datasets (S4–S6 Tables). Quantity disagreement made up the majority (9.1% of the overall 10.6%) of the overall disagreement between LCMAP and Lark *et al.*; the remaining disagreement comes from allocation disagreement. This indicates that the majority of disagreement between LCMAP and Lark *et al.* stems from definitional differences between the datasets, rather than differences in the spatial allocation of cropland and non-cropland. We further observe (S4 Table) that a large subset of the area that LCMAP classifies as stable cropland is classified by Lark *et al.* as stable noncropland (161 million acres), as expected from the more expansive definition of cropland in the LCMAP (Table 2). The reverse case—area where LCMAP reports stable noncropland while Lark *et al.* 2020 reports stable cropland—is much smaller (5.8 million acres). Similarly, we observe that the case

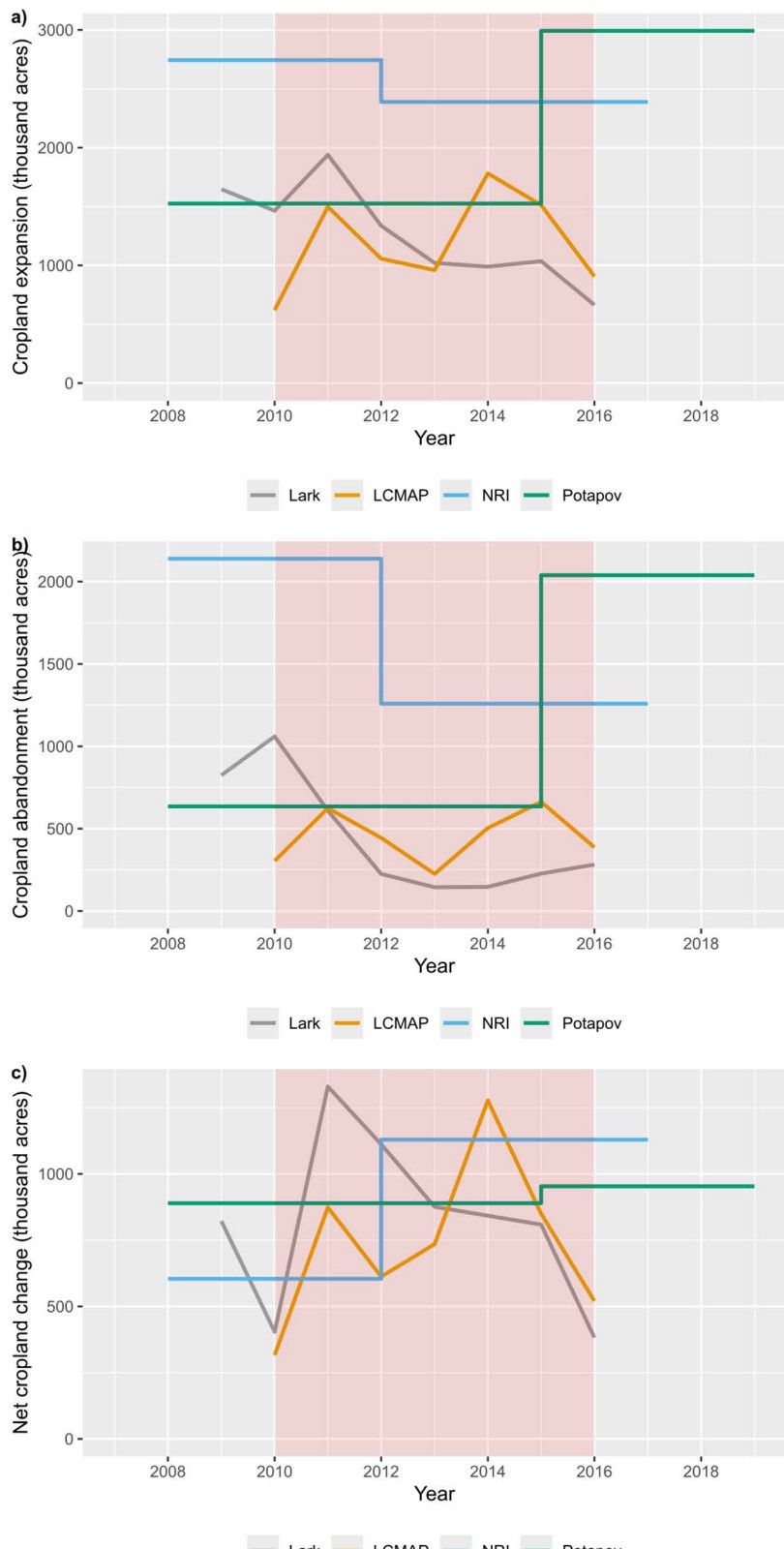

**Fig 4. Reported cropland abandonment, expansion, and net change over time across all datasets.** This figure shows the quantity (in thousands of acres) of change between 2008 and 2017 for cropland expansion (a), cropland abandonment (b), and net change (c).

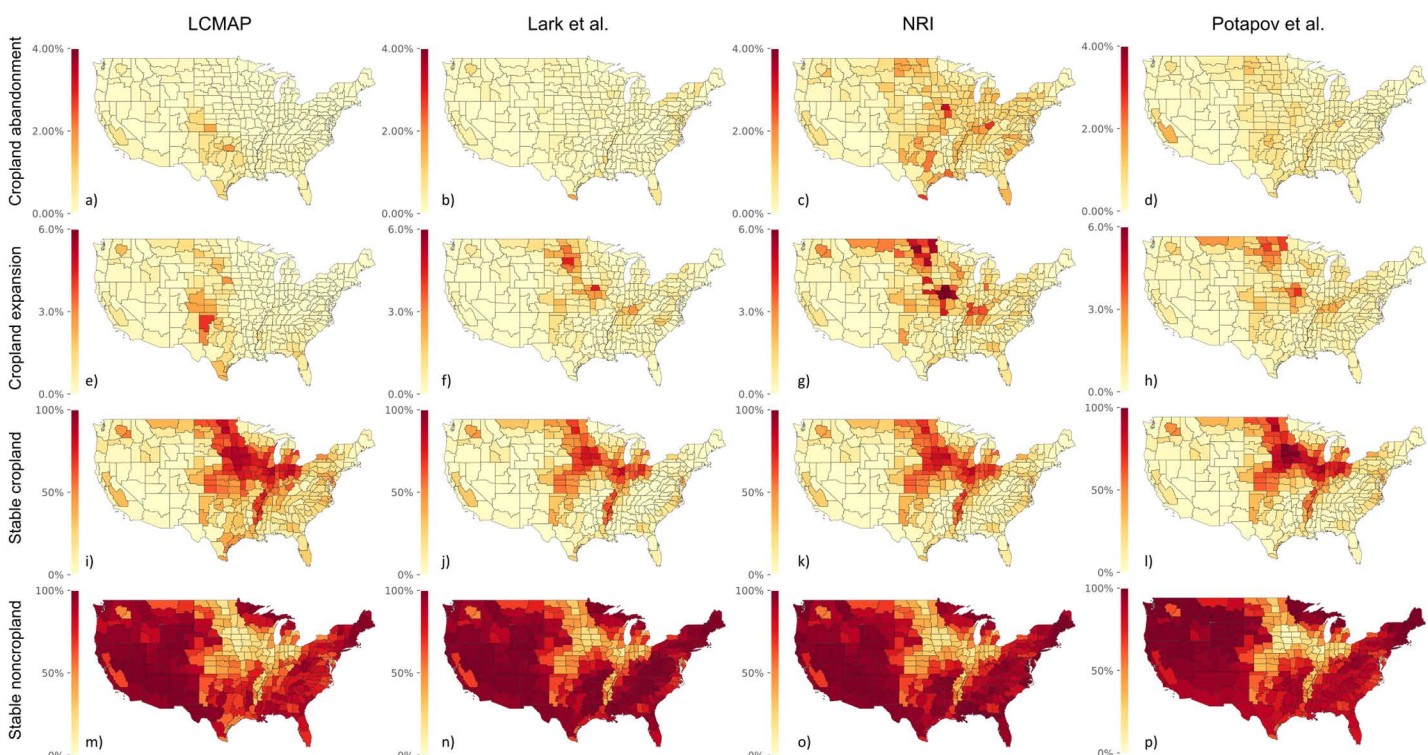

**Fig 5. Overview for reported LTPC quantity (expressed in terms of percent of total CRD area) from 2010-2017 for each dataset.** While data from LCMAP, Lark *et al.*, and Potapov *et al.* are published as raster files with a spatial resolution of 30 square meters, we aggregate by CRD for the purposes of comparing with the NRI data and for ease of displaying and interpretation. Note that the scale of the first two rows ranges from 0-6% of CRD area, while the scale of the second two rows ranges from 0-100% of CRD area. See S4 Table for absolute acreage by CRD.

where LCMAP reports stable cropland and Lark *et al.* 2020 simultaneously reports cropland expansion (5.2 million acres) is much more common than the reverse, where LCMAP reports cropland expansion and Lark *et al.* 2020 reports stable cropland (1.2 million acres).

In contrast to overall disagreement, when considering only cropland abandonment and cropland expansion, we observe that the allocation disagreement outweighs the quantity disagreement. This indicates that most of the disagreement between LCMAP and Lark *et al.* for these classes is the result of differences in spatial allocation (the placement of individual pixels). We calculated a Kappa coefficient ($\kappa$) of 0.69 indicating fair pixelwise agreement between land cover change derived from the LCMAP product and land cover change reported by Lark *et al.*

## Discussion

We compared the spatial and temporal distribution of cropland and cropland changes across CONUS from 2010-2017, derived from four land cover products—LCMAP, NRI, Lark *et al.* (derived from CDL in Lark *et al.* 2020), and Potapov *et al.* (from Potapov *et al.* 2022). These datasets vary in their source information, spatial scale of representation, frequency of data collection and reporting, land class definitions, categorical detail, and time intervals of representation. To make a direct comparison we harmonized—to the extent possible—spatial scales, processing steps, and time periods of consideration, as described in our methods. We made no attempt to harmonize the definitions beyond a simple cropland/noncropland binary as this would require re-deriving the original datasets and part of the objective of this study

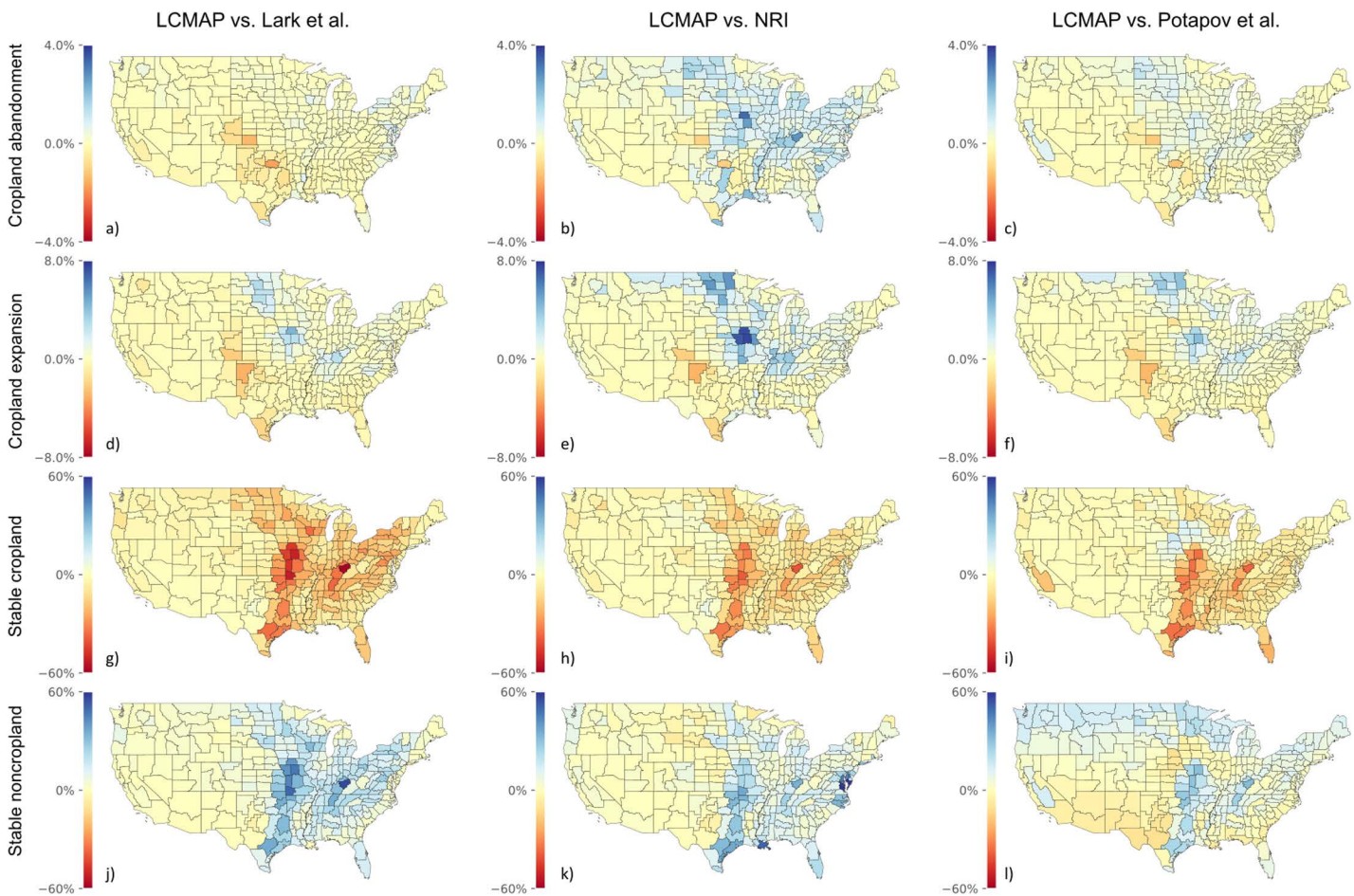

**Fig 6. Overview for LCMAP comparisons with Lark** *et al.***, Potapov** *et al.***, and NRI broken down by CRD.** Each subplot shows the signed value of the difference between reported values expressed as a percentage of the CRD area. Negative (red) numbers indicate that LCMAP reported higher quantities of a category, while positive numbers (blue) indicate that the other dataset reported higher quantities. Note the differences in unit scaling between the rows.

was to understand the different patterns of cropland expansion given the definitions embedded in the datasets. For comparison to data products that are available annually at pixel scales we used the methods described in Lark *et al.* 2020 for spatial filtering and assigning transition classes to stable, expansion, abandonment, or intermittent categories. For comparisons to data products that are available only at larger spatial scales (NRI) or longer time frames (NRI and Potapov *et al.*), we aggregated annual pixel-scale data both spatially and temporally.

Definitional differences in land cover classes between datasets (Table 2, S1 Table) appear to drive a large portion of the overall differences in reported quantities and spatial distribution of change. This definitional discrepancy results in differences between estimates of cropland extent and cropland change, as well as differences in how cropland dynamics manifest at local and regional scales.

We attribute the differences in quantities of both the stable classes (stable cropland and stable noncropland) and dynamic classes (cropland expansion and cropland abandonment) to this disparity. Cropland as defined by LCMAP encompasses many types of managed grassland and pastureland (Table 2), such that transitions between cultivated crops and managed grasses would not be captured as cropland expansion or abandonment in the LCMAP but would instead be reported as stable cropland. This contributes to the lower rates of abandonment

and expansion reported in LCMAP and its higher amount of stable cropland (and lower noncropland) relative to the other datasets evaluated. As a result, cropland expansion in the LCMAP is likely a conservative estimate compared with other data sources. Put another way, cropland expansion in LCMAP may represent conversion of "more natural" lands than cropland expansion from Lark *et al.*, NRI, or Potapov *et al.*, which would include varying amounts of pasture and hay. Whether this is desirable or undesirable depends on the research question at hand.

The other three datasets capture at least some of the conversions between cropland and managed pastureland and grassland. The cropland definition from Lark *et al.* excludes all grassland and pastureland and should capture the greatest number of transitions between cultivated crops and various types of grasslands. Cropland definitions from the NRI and Potapov *et al.* include only some unmanaged pastureland and should capture an intermediate level of cropland-grassland transitions. Reflecting these definitional differences, the NRI and Potapov *et al.* data report quantities of stable cropland and noncropland that are in between the LCMAP and Lark *et al.* estimates. Despite the intermediate estimates for stable classes, the NRI and Potapov *et al.* reported much higher overall levels of dynamic classes. We assume the estimates from the NRI are likely the most reliable, suggesting that there is more expansion and abandonment occurring than the LCMAP or Lark *et al.* suggest. This could be from the processing steps applied to the LCMAP and Lark *et al.* which may remove or average over some expansion and abandonment that is occurring. Alternatively, some portion of the expansion and abandonment in Potapov *et al.* and the NRI may actually fall into the "intermittent" class, which is not possible to estimate with those datasets. Regardless, it was interesting to note that many of these differences canceled out with estimates of net cropland expansion from all four datasets being remarkably similar.

For the spatial patterns of the dynamic classes (cropland expansion and cropland abandonment), however, we see large disagreements, especially in northern Texas and across the Midwest. NRI, Lark *et al.* 2020, and Potapov *et al.* 2022 report the majority of cropland expansion in the Dakotas, Minnesota, and Missouri, where a large portion of the change appears to be from CRP, pasture, and non-alfalfa hay being converted to row crops, which the LCMAP misses. In contrast, LCMAP reports most cropland expansion in Oklahoma and northern Texas, where conversion reflects more natural grasslands and rangelands being converted into row crops. Given the LCMAP's broader categorization of cropland, it may also be capturing and including grassland and rangeland conversion to more intensively managed pasture and hay. The fact that some of these transitions also show up in Lark *et al.* (Fig 6f), though lower in magnitude compared with LCMAP, suggest that at least some of this conversion is to row crops. Further research into the CDCC algorithm underlying the LCMAP data and its treatment of cultivated pasture and hay could help explain these differences in spatial distribution.

Stehman *et al.* (2021) provided a robust assessment of LCMAP for 1985-2017. That study, however, is mainly focused on the accuracy of the LCMAP with the associated 25,000 reference samples across all land cover classes compared with the NLCD, rather than the particular cropland change classes of interest here. Stehman *et al.* found that the user and producer accuracies for cropland in the LCMAP were 70% and 93%, respectively, while in the NLCD they were 89% and 86%. They also found that user and producer accuracies for grassland in the LCMAP were 88% and 80%, respectively, while in the NLCD (the basis of grassland in the CDL) they were 65% and 67%.

Unfortunately, the LCMAP reference data which is intended to support validation is not sufficient to support validation of cropland expansion or abandonment. This is in part because these reference data are spread over a long time series (from 1984 to 2016) resulting in just over 750 reference points per year. In addition, the relative rarity of cropland expansion and

abandonment (relative to stable cropland and stable non-cropland) at the level of the CONUS means that the density of reference points may not be adequate to gauge the accuracy of these rare transition classes. This does not mean that such conversions are not occurring, but it does mean that the spatial accuracy of these allocations are hard to assess with current tools and the existing reference data. A more stratified validation dataset based on NRI data points or some other dataset would be more useful in validating the accuracy of the dynamic classes.

Temporally, we see that LCMAP identifies a greater proportion of cropland expansion in the latter part of our study period relative to Lark *et al.* 2020. Given this, and the differences in definitions, this may suggest a larger degree of cropland expansion into pasture and hay during the earlier portion of the period examined (which is missed by LCMAP) followed by greater cropland expansion into natural grasslands and rangeland during the latter portion of the period examined (potentially accompanied by additional conversion of grassland/range-lands to pasture/hay). Such a trend would be consistent with economic theory and models [27] which suggest that pasture and hay lands are more responsive and would be the first to convert following the crop prices and demand spike between 2008 and 2012, and that those would be followed by more durable grasslands being converted later.

LCMAP's use of the CDCC algorithm to detect change and suppress temporal noise is another factor that may affect the LCMAP's tendency to report lower amounts of cropland expansion and abandonment in comparison to the other datasets. This interpretation is supported by the difference in the reported intermittent cropland acreage between Lark *et al.* and LCMAP. LCMAP reports a much lower quantity of intermittent cropland than Lark *et al.*, despite drawing from the same methodology to assign long-term pattern classes. This likely reflects both the increased noise in the raw annual CDL data underlying Lark *et al.* and the use of the intermittent cropland category of Lark *et al.* to capture and remove not only noise-induced misclassification errors but also crop and noncrop rotations from its estimates of cropland expansion and abandonment. This finding suggests that the unprocessed LCMAP data is more appropriate as a base land cover map for estimating change than unprocessed CDL, as LCMAP's continuous availability and the use of the CDCC algorithm both serve to mitigate the impact of single-year misclassification errors [20], similar to the effects of the Lark *et al.* postprocessing on the CDL.

Collectively, we may interpret this combination of spatial and temporal results to mean that there is a greater proportion of cropland expansion from intensively managed grasslands in the Dakotas, Minnesota, Iowa, and Missouri, particularly during the early part of our study period. However, there is also cropland expansion from natural grasslands throughout the period and across the nation, but this type of conversion is particularly concentrated in Texas and Oklahoma during the latter part of the study period.

Overall, we find that LCMAP reports a greater extent of stable cropland and lower extent of stable noncropland in all comparisons, due to its relatively more expansive definition of cropland. We see the greatest disagreement between LCMAP and the other three datasets in Texas and the Midwest (Fig 6). The inclusion of hay and managed pasturelands in the LCMAP definition of cropland serves to mask some grassland-to-cropland and cropland-to-grassland transitions—transitions which are flagged as conversions (to differing extents) by other data-sets in our comparison. As a result of this expansive definition of cropland, combined with the general ambiguity in the definitions of pastureland, hayland, grassland, and rangeland (which is not a problem unique to LCMAP), the LCMAP may miss an entire set of transitions that may be environmentally relevant (Table 6).

Our findings highlight important considerations for using LCMAP data in subsequent analyses of U.S. cropland dynamics. In particular, applications focused on soil carbon where conversion of managed pasture may be important to include should consider using the

**Table 6. Potential sources of disagreement and effects on change estimates.**

| Cause of disagreement | Predicted effect on change estimates | Expected effect on LCMAP comparison |
|---|---|---|
| Inclusion of grassland and pastureland in LCMAP definition of cropland | Reduced or nonexistent cropland-to-grassland abandonment and grassland-to-cropland expansion<br>Increased stable cropland | Reduced expansion and abandonment in LCMAP compared to other datasets<br>Reduced stable noncropland in LCMAP compared to other datasets<br>Increased stable cropland in LCMAP compared to other datasets |
| LCMAP use of CDCC algorithm to determine time of spectral shift over multiple years | Reduced temporal noise (fewer spurious conversions) resulting in fewer pixels categorized as intermittent cropland or, for conversions in the final years of the study, abandonment/expansion | Reduced intermittent cropland in LCMAP compared to Lark *et al.* 2020<br>Slightly reduced cropland and expansion in LCMAP compared to Lark *et al.* 2020 |
| Aggregation methods for products with multi-year intervals (NRI and Potapov *et al.* 2022) | Potential double-counting of expansion and abandonment in subsequent years for multi-year interval datasets | Reduced expansion and abandonment in land change maps derived from annual products (LCMAP and Lark *et al.* 2020) compared to these other datasets (NRI and Potapov *et al.* 2022) |

processed CDL (as in Lark *et al.* 2020), NRI, Potapov *et al.* 2022, or some other data source that includes that as a conversion, while applications focused on habitat for threatened and endangered species may consider using the LCMAP which focuses on conversion of more natural land cover types. Thus, applications that require detailed categorical differentiation between cropland and grassland, or capture of conversions between cropland and managed grasslands, should look into alternative datasets or ways to augment LCMAP data. Applications that examine the impact of cropland broadly may therefore be able to take advantage of the benefits of LCMAP—its long historical record, frequency of releases, and integrated change detection approach—without detriment to the overall aim of the assessment. Regardless, researchers and policymakers need to be acutely aware of the underlying strengths and limitations of these datasets when potentially employing them for specific research questions or in crafting policies.

It is important to note that although our focus was on comparing land use change estimates from 2010-2017, these findings have implications beyond that window. Cropland expansion and abandonment are long term phenomena in the U.S. influenced by many factors; and, as noted in the introduction, by the first year of national CDL availability (2008) the growth of corn ethanol consumption in the U.S. was roughly halfway finished. Thus, longer timeseries data are helpful for assessing longer term patterns in the U.S., especially for biofuels. Various national datasets, including the NRI, show a long term decline in cropland in the U.S. beginning in the 1980s, a trend that reversed sometime in the mid-2000s. The LCMAP confirms this reversal, with net cropland losses from 2000-2008 (S3 Fig) switching to net cropland gains from 2010-2017 (S2 Fig). Thus, the longer timeseries of the LCMAP is useful when examining the potential implications of these longer trends in land use change from biofuels and other factors.

We also note that the steps we used to harmonize the data in this study provides a roadmap for the responsible comparison of land cover data. An important step is the translation of annually available land cover data into land cover transition classes, taking full advantage of the availability of interim years [28]. In this study we mimic the rules outlined by Lark *et al.* 2020 to classify pixels based on their temporal trajectories into stable, expansion, abandonment, or intermittent classes. We note however that this is a subjective approach, application- and dataset-specific, and one that could potentially have a significant impact on the resulting output. As releases of high-resolution land cover data products become more frequent [29], guidance for common approaches to trajectory classification will help the scientific community harmonize efforts to better understand land cover trends [30].

Our approach also demonstrates the importance of comparison among land cover datasets to inform where we have confidence in use of these data and where caution and additional research may be needed. As the abundance, resolution, and availability of earth observation-based land cover data are improving rapidly, there will be an ongoing need to improve the science and guidance for comparing among datasets. Our approach builds on investigations of the agreement between global land cover datasets from, for example, Hua *et al.* 2018, Zhang *et al.* 2021 and Venter *et al.* 2022, while adding an important assessment of differences in land cover change [31,32].

Our assessment highlights improved availability and accessibility of validation data as another key requirement that would support more informed and responsible use of land cover and land cover change datasets [30]. The comparisons presented here do not demonstrate inaccuracies in these data products (although those do exist). Instead, they highlight differences and similarities, which may be due in large part to definitional discrepancies and other differences in the construction of the data.

As noted in the results, the usefulness of the LCMAP validation data is limited in terms of the focus of this effort. More than 90% of reference pixels fell into stable classes (stable cropland and stable non-cropland). Further development of an LCMAP reference dataset that captures the dynamic classes (cropland expansion and cropland abandonment) would be helpful to better evaluate the accuracy of land cover change products; however, this falls outside the scope of the current analysis. We also recommend that the LCMAP developers consider partitioning some portion of the managed pasture/grassland into either grassland or into its own category, if possible, since many researchers and policymakers use these data for analysis of environmental impacts and conversion of these land cover types to row crops can result in significant environmental damages.

Despite the differences in type and location, it is important to acknowledge that all of these datasets report cropland expansion in CONUS. This was reported in EPA's Second Triennial Report to Congress on Biofuels, which found that once definitional differences were harmonized (to the degree possible) among the NRI, CDL-based studies, the US Agricultural Census, and two other sources, all of these datasets yielded similar estimates of cropland expansion (3-7.8 billion acres from 2007/08 and 2012) [19]. We also found surprising agreement among datasets in terms of net cropland expansion, even though the type and spatial location varied.

Examining the consistency of cropland area and change estimates across a number of datasets provides valuable insight into areas of agreement and disagreement, meaningful reasons for observed differences, and guidance for selecting data for a desired application. This will become increasingly important as satellite imagery-based data becomes more abundant, accessible, and spatially and temporally detailed. Indeed, the challenge of understanding trends in land cover change and agricultural expansion and contraction is rapidly evolving from one of *not enough* data to one of *too much* data that are not easily compared, and which produce seemingly contradictory results. Studies like this one, that grapple with variations in class definitions, time periods of reporting, spatial resolution, and other methodological differences, are increasingly needed to enable decision makers to use land cover data with greater confidence.

## Supporting information

**S1 Table. Land cover class definitions by dataset.** Note that these definitions apply to the underlying land cover maps used to produce land change estimates.
(DOCX)

**S2 Table. LCMAP land cover definitions.**
(DOCX)

**S3 Table. Expanded cropland class definitions across datasets.**
(DOCX)

**S4 Table. Pixelwise long-term pattern class agreement between LCMAP and Lark et al. 2020 in thousands of acres from 2008 - 2017.**
(DOCX)

**S5 Table. Kappa coefficient determination of agreement between LCMAP and Lark et al. 2020.**
(DOCX)

**S6 Table. Agreement measures for comparison between LCMAP and both Lark et al. 2020.** For the cropland expansion and cropland abandonment disagreement, percentages indicate the fraction of all pixels in the entire dataset. For this reason, we do not calculate overall agreement or disagreement attributable to cropland expansion and cropland abandonment. Overall agreement indicates the extent to which pixelwise values between these datasets agree; overall disagreement indicates the extent to which pixelwise values disagree. Quantity and allocation disagreement together make up the overall disagreement, with quantity disagreement indicating the portion of the overall disagreement attributable to differences in the number of pixels each dataset assigns to each class.
(DOCX)

**S1 Fig. Spatial comparison of stable cropland and cropland expansion between LCMAP and Lark _et al._** The first column of this figure shows the spatial distribution of pixels where LCMAP reported stable cropland and Lark _et al._ reported cropland expansion. The second column shows the reciprocal relationship, where LCMAP reported cropland expansion and Lark et al. 2020 reported stable cropland. Note that each subplot has a different scale and that the first row is in terms of the percent of the CRD area, while the second is in thousands of acres.
(TIF)

**S2 Fig. LCMAP reported cropland expansion and abandonment (2010-2017).**
(TIF)

**S3 Fig. LCMAP reported cropland expansion and abandonment (1998-2009).** This figure shows the quantity (in thousands of acres) of cropland expansion and abandonment between 1998 and 2009. Because of how we define long-term pattern classes (Table 3), for the 1998 – 2009 interval, the first potential year of a class change is 2000 and the last is 2008. We performed this analysis separately from the main analysis described earlier in the paper using the same methods.
(TIF)

## Acknowledgments

This manuscript was developed under Contract 68HERD20A0004/68HERH22F0010 with the EPA's Office of Research and Development. The views presented are those of the authors and do not necessarily represent the views or policies of the US Environmental Protection Agency. We thank Steven LeDuc and Jesse Miller for their thoughtful comments on an earlier version of the manuscript. Tyler Lark contributed data based upon work supported in part by the Great Lakes Bioenergy Research Center, U.S. Department of Energy, Office of Science, Biological and Environmental Research Program under Award Number DE-SC0018409.

## Author contributions

**Conceptualization:** Gray Martin, Kemen Austin, Tyler Lark, Christopher M. Clark.

**Data curation:** Gray Martin.

**Investigation:** Gray Martin.

**Methodology:** Gray Martin.

**Software:** Gray Martin.

**Writing – original draft:** Gray Martin.

**Writing – review & editing:** Gray Martin, Kemen Austin, Tyler Lark, Stanley Lee, Christopher M. Clark.

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
