## [Decision Letter · Decision Letter 0]

7 Aug 2024

PONE-D-24-19267Tracking cropland transitions: a comparative analysis of U.S. land cover change dataPLOS ONE

Dear Dr. Martin,

Thank you for submitting your manuscript to PLOS ONE. After careful consideration, we feel that it has merit but does not fully meet PLOS ONE’s publication criteria as it currently stands. Therefore, we invite you to submit a revised version of the manuscript that addresses the points raised during the review process.

Two reviewers have evaluated your article and they both feel it still needs to be revised before it can be considered again for publication in PLOSONE.

The first reviewer had positive feedback on the quality and relevance of the paper, and recommended a minor revision incorporating a few errors he has highlighted on certain aspects of wording and language in various sections of the text. He also advices the authors to take into consideration the fact that various imagery data sources have inherently different spatial resolution levels, and that needs to be taken into account when making comparative inferences.

The second reviewer had more profound concerns about the technical soundness and organization structure as well as general presentation of the articles and, although did not recommend rejection, suggested much more in-depth modifications in almost all sections. In particular, the reviewer points out that the article appears to be primarily based on the Lack et al (2020) dataset yet the authors claim it to be founded on the Cropland Data Layer dataset. This needs to be clarified and the write-up, including all discussion and conclusion amended accordingly.

There is also need to account for the temporal aspects including overlaps (annual, semi-annual etc.)  as well as longevities of the datasets sources (LCMAP, Lark et al; NRI etc.) before appropriate comparisons may be made and discussed with accuracy

It must be adequately justified why the NRI and Potapov datasets were included in the article since they received much less treatment, or appear to be not as central to the comparisons and discussion as are the LCMAP and Lark et al components. Similarly, NAIP and NLCD datasets seem to be only of peripheral significance to the study

Material in Discussion and Results sections must not be allowed to overlap/spill over into each other

Be sure to add line numbers throughout the text pages

Reproduce illustrations with more clearly distinct reference information (legends and captions) making sure to make figure more explicitly color-code separated, remembering that some readers are color-blind. Also avoid footnotes throughout.

To reduce overall clutter and number of illustrations, try to merge and collapse closely related tables together, and try to consolidate figures presenting related information into separate panels of same figures if possible

Follow the author guidelines on the formatting style of PLOSONE to the letter, including rules for formatting tables and figures as well as reference listing

We look forward to receiving your revised manuscript.

Kind regards,

Nickson E. Otieno

Academic Editor

PLOS ONE

 [RTI received financial support for this work from EPA’s Office of Research and Development (https://www.epa.gov/aboutepa/about-office-research-and-development-ord) under contract 68HERD20A0004/68HERH22F0010. The views presented are those of the authors and do not necessarily represent the views or policies of the US Environmental Protection Agency.].  

3. We note that Figures 7 and 9 in your submission contain [map/satellite] images which may be copyrighted. All PLOS content is published under the Creative Commons Attribution License (CC BY 4.0), which means that the manuscript, images, and Supporting Information files will be freely available online, and any third party is permitted to access, download, copy, distribute, and use these materials in any way, even commercially, with proper attribution. For these reasons, we cannot publish previously copyrighted maps or satellite images created using proprietary data, such as Google software (Google Maps, Street View, and Earth). For more information, see our copyright guidelines: http://journals.plos.org/plosone/s/licenses-and-copyright.

1. You may seek permission from the original copyright holder of Figures 7 and 9  to publish the content specifically under the CC BY 4.0 license.  

Additional Editor Comments (if provided):

Reviewers' comments:

Reviewer's Responses to Questions

**Comments to the Author**

1. Is the manuscript technically sound, and do the data support the conclusions?

Reviewer #1: Yes

Reviewer #2: No

2. Has the statistical analysis been performed appropriately and rigorously? 

Reviewer #1: Yes

Reviewer #2: No

3. Have the authors made all data underlying the findings in their manuscript fully available?

Reviewer #1: Yes

Reviewer #2: No

4. Is the manuscript presented in an intelligible fashion and written in standard English?

Reviewer #1: Yes

Reviewer #2: Yes

5. Review Comments to the Author

Reviewer #1: Section Page # Para # Line # Comment/suggestion/question

Abstract 1 2 1 “……. fuel put pressure on ……..” instead of “….fuel influence the ……”

Introduction 3 3 2 Check that “frequency of data collection” and “time intervals of representation” are not used in the same context?

4 3 “…….reasons for these …….”

4 Table 1 For Global Cropland Expansion in the 21st Century (Potapov), the data is Landsat GLAD ARD and not MODIS NPP? Check https://nature.com/articles/s43016-021-00429-z and read the section “Methods”. Also check the interval, which is four year and not semi annual?

5 1 9 …….to understand the ………..

2 1 “………for the conterminous…”

2. Methods 7 1 1 ‘…….product.for the CONUS based on time-series…….”

10 1 What proportion this would be of the total cropland area? Check results for this.

Figure 2 It is confusing and hard to read and understand. May be re-structured for better understanding.

18 1 4 What about conversion to other land covers such as built up area and shrublands?

Table 6 Include area unit (000 acres) in column one. E.g. Cropland abandonment (000 acres)

In Potapov et al., Will adding managed pastures to the stable croplands from non cropland, be not closer to the LCMAP estimates? Check that. This will mean a definition variation/error.

27 1 6 “…….extent possible – definitions……. How?

2 5 Check that managed grasslands are included in the croplands in Potapov et al.?

28 3 1 Quantities or “extent” in “………higher quantities of stable ….”

General comments The comparison is not at one spatial scale, as various imagery sources have different spatial resolution. How this spatial resolution in the range of 30m to 0.5 meter was addressed for uniformity?

It would have been more appropriate to have used independent reference data for accuracy check of the various products, and then compared those among each other and with that of LCMAP. The review article projects that LCMAP is used as a reference. However, considering that LCMAP will have commission and omission errors, and will certainly needs an independent dataset to check with.

Reviewer #2: The study assessed factors that drive changes in cropland estimation between Land Change Monitoring, Assessment, and Projection (LCMAP), National Resources Inventory (NRI), Cropland Data Layer (CDL, Lark et al. 2020), Potapov et al. (2022) datasets across coterminous United States. The study is interesting and important because of the various land cover datasets that exist and how differences between them could influence the outcome of their applications.

The study has several issues firstly with the “Cropland Data Layer”. The study does not use the Cropland Data Layer but the Lark et al 2020 dataset which is based on the Cropland Data Layer and National Land Cover Dataset. In some sections of the text it is mentioned as Cropland Data Layer e.g., the Abstract and Table 1 but in others as Lark et al 2020 dataset. It would be more appropriate to use just say dataset from Lark et al 2020 or something similar throughout the text just as the study states “dataset from Potapov et al. (2022)”.

The major issue is with the harmonisation or lack of with the cropland definitions and temporal scale of the datasets. Without the homogenisation of the cropland definitions and carrying out a sensitivity analysis it is not possible to say if the cropland estimation is based on the differences in definitions or some other factor. The temporal differences of the datasets raises the most concerns in the study. The LCMAP and Lark datasets are annual datasets and have a partial temporal overlap. The NRI dataset exists only for 2007, 2012, and 2017 and while 2007 is included in the analysis the LCMAP and Lark are from 2008 onwards. This could greatly influence the results or not since it's a 1 year difference but the study needs to account for this. The Potapov dataset used covers 2019 whereas the LCMAP, Lark, NRI datasets are till 2017. Again the study does not account for how this difference could/could not influence the results.

If the objective of the study is to compare different datasets then they would have to be harmonised to be comparable otherwise it raises inconsistencies and would not answer the aim of the study “The aim of this study is to understand how estimates of cropland change from LCMAP differ from estimates from other data products across a common time period (2008-2017) and which factors drive these differences.”

The Results section reads more like a discussion while the Discussion reads like the results. These two sections need major improvements.

Overall the study seems to be more of a comparison between LCMAP and Lark datasets with NRI and Potapov datasets being forced into the comparison.

The comments about specific issues are given below. Given the lack of line numbers, the page and paragraph numbers are used instead

Martin 1: “National Resources Inventory (NRI), (2) the Cropland Data Layer (CDL, Lark et al. 2020), and (3) a dataset from …” Here, CDL is within the () whereas everywhere else in the text the short forms are not included within the reference brackets e.g., Martin 3, paragraph 3: “the USGS National Land Cover Database (NLCD) (Homer 2020), the USDA Cropland Data Layer (CDL) (US Department of Agriculture 2021a) …”.

Martin 1: “LCMAP reports more stable cropland and less stable noncropland in all comparisons, likely due to its expansive definition of “cropland” including managed grasslands (pasture and hay),...” how can this be stated when the differences in the temporal scales of the datasets have not been controlled?

Martin 1: “We found that most of the pixel-level disagreements (86%) were due to definitional differences among datasets, whereas the remainder (14%) were from a variety of causes including spatial allocation.” this gives the impression that the comparison is between all the four datasets when that is clearly not the case.

Martin 1: “This study also includes a brief assessment of land use change from 2000-2008, an often-overlooked period, and find that many of the studies focused on 2008 and after may omit large amounts of cropland expansion and abandonment.” I fail to see the point of this assessment, it would be better to focus on the main findings in the abstract.

Martin 3: paragraph 3 “These tools vary in their…” should it not be datasets instead of tools since datasets is generally used across the text?

Martin 3: paragraph 3 “USDA NASS annual surveys reported an increase in principal crop planted

area of 0.38 million hectares (US Department of Agriculture 2021b), while a study based on the CDL data reported a value of net cropland expansion nearly four times larger” is “principal crop planted

area” defined the same as “net cropland expansion”? Or do they mean different things and that's why there is a difference.

Martin 3: paragraph 3 “several of these datasets (e.g., NRI, CDL, Census, etc.) yielded similar results in” is etc. a dataset? what is the Census dataset?

Martin 4: Table 1 - The inclusion of National Agricultural Imagery Program (NAIP) and National Land Cover Database (NLCD) seems unnecessary. There appears to be footnotes below the table which are not permitted - https://journals.plos.org/plosone/s/submission-guidelines. Additionally it might be more appropriate to move this to the “a. Data description” section.

Martin 4: Table 1 - While Table 1 “Availability” states that Potapov dataset is “Semiannual (2000-2019)” the website https://glad.umd.edu/dataset/croplands indicates the the dataset is available at “four-year intervals (2000-2003, 2004-2007, 2008-2011, 2012-2015, and 2016-2019)”. Is the dataset available at semi annual and four year intervals?

Martin 5: paragraph 2 “In this study we compare LCMAP-derived … not be an independent comparison.” this entire paragraph reads as what was done in the study and would be better to include it in “the aim of the study” paragraph.

Martin 5: paragraph 3 “Stehman et al. (2021) provides a robust assessment … while in the NLCD they were 65% and 67%” this section should be its own separate section and before the “the aim of the study” paragraph. Additionally, are there no other studies that compared land cover between different datasets? does not have to be limited to the US datasets/studies.

Martin 7: “a. Data description” the heading is data description but only the LCMAP data is described. What about the other datasets?

Martin 7: paragraph 1 “Although LCMAP has only been available since 2020, this dataset has found use across a range of applications (Healey and Rover 2022) (Xian et al. 2021) (Tollerud, Brown, and Loveland 2020).” All the references should be within one bracket ().

Martin 8: Table 2 - it would be better to keep this in the supplementary.

Martin 9 and 10: Table 3 and Figure 1 - Figure 1 is a good visual explanation. The contents of Table 3 can be included in Figure 1 to make things more concise since the two explain the same things.

Martin 10: Again, footnotes are not permitted.

Martin 10: “c. Definitional differences” it would make more sense to have this section after “a. Data description”. This way cropland and non-cropland are described before talking about the long-term pattern class definitions.

Martin 11: paragraph 1 “(“National Resources Inventory Glossary | NRCS” n.d.)” this seems like an odd reference style.

Martin 11: Table 4 - Is there a reason “permanent managed pasture” is not included in cropland in the Potapov dataset when this subclass is included in cropland in LCMAP dataset? Or why “perennial woody crops including orchards and vineyards” is not included in cropland in the Potapov dataset when this subclass is included in cropland in Lark dataset? There’s mention of homogenisation in the discussion section Martin 27 paragraph 1 “To make a direct comparison we therefore harmonised—to the extent possible …” but there does not seem to be any homogenisation of the classes done. How would the results of the study change if the definitions are harmonised? Isn’t this what the study aims to do?

Martin 11 and 12: Table 4 and Figure 2 - Figure 2 is a good visual explanation. Again, the contents of Table 4 can be included in Figure 2 to make things more concise since the two explain the same things. Additionally, consider using hollow coloured boundaries to make things easier to read and add categories in the legend for the solid and dash black blocks.

Martin 12: “d. Aligning temporal differences” - How can the use of the 2007 NRI dataset be justified when it does not temporally overlap with the other datasets used? How can the use of the 2019 (2017-2019) Potapov dataset be justified when the other dataset used are just up to 2017? The abstract states “LCMAP reports more stable cropland and less stable noncropland in all comparisons, likely due to its expansive definition of “cropland” including managed grasslands (pasture and hay), though overall agreement remains high” this could also be due to the temporal scale of the datasets?

Martin 16: Figure 4 - People who are colour-blind will have a hard time reading this. Consider using a different colour palette. Also, add the full form for “LTPC”.

Martin 16: paragraph 1 “compared with other sources, most evident in the comparison to Lark et al.

2020. We can attribute some … LCMAP estimates than estimates from Lark et al. 2020.” the Result section should state the results of this study, not compare it to the findings of other studies or why the results have a certain pattern, that's what the discussion section is for.

Also what would the estimates look like when the definitions are harmonised?

Martin 17: Table 6 - this provides an explanation of the results, it should be as a text in the discussion section.

Martin 17-20: most of the text describes the meaning of the results which is not what this Result section is for.

Martin 18: Table 6 - Should be table 7 and should be the same style as the other tables.

Martin 21: Figure 5 - What is the point of this figure when the same is plotted in Figure 6?

Martin 21: Again, footnotes are not permitted.

Martin 22: Figure 6 - captions are at the top of the figure. They should be at the bottom.

Martin 25: Figure 7 - People who are colour-blind will have a hard time reading this. The labels on the right are externally added and not aligned with images and are of higher resolution than other parts of the image.

Martin 26: Figure 8 - Should be figure 8. Again, people who are colour-blind will have a hard time reading this. The labels on the right are externally added and not aligned with images and are of higher resolution than other parts of the image.

Martin 27: The Discussion section from Martin 27 paragraph 1 to Martin 29 paragraph 1 is easy to follow and makes sense after that it seems to go on a tangent. Martin 29 paragraph 2 “The longer temporal record of … application in question.” why is this aspect about the temporal interval included here when the paragraph above talks about the temporal aspects.

Martin 29: paragraph 3 “rules outlined by Lark et al. 2017 to classify pixels based on their temporal

trajectories into stable, expansion, abandonment, or intermittent classes. We note however that this is a

subjective approach, application- and dataset-specific, and one that could potentially have a significant

impact on the resulting output.” with this and the lack of harmonised definitions and time interval does the study even assess what factors are influencing the estimates?

Martin 34-35: Supplementary Figure 1 and 2 - People who are colour-blind will have a hard time reading this. The labels on the right are externally added and not aligned with images and of higher resolution than other parts of the image.

6. PLOS authors have the option to publish the peer review history of their article (what does this mean? ). If published, this will include your full peer review and any attached files.

**Do you want your identity to be public for this peer review?** For information about this choice, including consent withdrawal, please see our Privacy Policy .

Reviewer #1: No

Reviewer #2: **Yes: ** Vishesh L. Diengdoh

---

## [Author Response · Author response to Decision Letter 1]

28 Oct 2024

This is included in the attached files with correct text coloring:

Thank you for the opportunity to review the manuscript. We feel that the reviewer comments were largely supportive and constructive. We have worked to respond to all of the comments (blue text) and explained our rationale below. Thank you again, and we look forward to hearing from you.

PONE-D-24-19267

Tracking cropland transitions: a comparative analysis of U.S. land cover change data

PLOS ONE

Dear Dr. Martin,

Thank you for submitting your manuscript to PLOS ONE. After careful consideration, we feel that it has merit but does not fully meet PLOS ONE’s publication criteria as it currently stands. Therefore, we invite you to submit a revised version of the manuscript that addresses the points raised during the review process.

Two reviewers have evaluated your article and they both feel it still needs to be revised before it can be considered again for publication in PLOSONE.

The first reviewer had positive feedback on the quality and relevance of the paper, and recommended a minor revision incorporating a few errors he has highlighted on certain aspects of wording and language in various sections of the text. He also advices the authors to take into consideration the fact that various imagery data sources have inherently different spatial resolution levels, and that needs to be taken into account when making comparative inferences.

We have revised according to these suggestions and detail those below.

The second reviewer had more profound concerns about the technical soundness and organization structure as well as general presentation of the articles and, although did not recommend rejection, suggested much more in-depth modifications in almost all sections. In particular, the reviewer points out that the article appears to be primarily based on the Lack et al (2020) dataset yet the authors claim it to be founded on the Cropland Data Layer dataset. This needs to be clarified and the write-up, including all discussion and conclusion amended accordingly.

We detail our responses below. Our paper is not primarily based on Lark et al. (2020), it is primarily based on the LCMAP, which we compare with three datasets, one of which is indeed Lark et al. (2020), which is based on the CDL. We have revised the paper to make it clear that we’re comparing our results with Lark et al. (2020), among others, and not the raw CDL. Apologies for the confusion.

There is also need to account for the temporal aspects including overlaps (annual, semi-annual etc.) as well as longevities of the datasets sources (LCMAP, Lark et al; NRI etc.) before appropriate comparisons may be made and discussed with accuracy.

We have re-written the methods to improve transparency and accuracy as this is precisely what we attempted to do but did not explain it adequately in the initial submission. Additionally, we ensured that the data in the figures and tables reflected a unified approach to temporal harmonization.

It must be adequately justified why the NRI and Potapov datasets were included in the article since they received much less treatment, or appear to be not as central to the comparisons and discussion as are the LCMAP and Lark et al components. Similarly, NAIP and NLCD datasets seem to be only of peripheral significance to the study.

This is clarified in the revisions. The Lark vs. LCMAP comparison appears to be central because the Lark estimates (2020) have been driving much of the debate and the discussion in the literature. So we felt it was appropriate to focus on that comparison. Additionally, given the similarities in both the underlying datasets (CDL and LCMAP) and the processing methodologies, we were able to perform a pixelwise comparison between Lark and LCMAP that we could not reasonably do for NRI and Potapov as well.

The NRI is generally considered the best overall estimate, but it is not a remotely sensed dataset and thus is not available at the pixel scale that the Lark et al. (2020) and LCMAP datasets are. Potapov is included because it is the most recent of the four estimates and because it has a high spatial resolution.

The NLCD is not included because it is not independent of the CDL (as explained in the original submission (page 5): “Furthermore, we omit NLCD because the CDL leverages NLCD for its non-crop classification and validation such that it would not be an independent comparison.”) NAIP is not included because it’s a repository of photographic images that have to be viewed one at a time (usually for validation purposes) and have not been stitched into a national or regional database to date to our knowledge. Thus, each has its reason for inclusion and omission, and we have fleshed that out more in the revisions by adding a section on dataset selection in the methods.

Material in Discussion and Results sections must not be allowed to overlap/spill over into each other.

We’ve revised the manuscript throughout to clean this up.

Be sure to add line numbers throughout the text pages.

We apologize for that omission; we have added page and line numbers.

Reproduce illustrations with more clearly distinct reference information (legends and captions) making sure to make figure more explicitly color-code separated, remembering that some readers are color-blind. Also avoid footnotes throughout.

We’ve cleaned up all the figures.

To reduce overall clutter and number of illustrations, try to merge and collapse closely related tables together, and try to consolidate figures presenting related information into separate panels of same figures if possible.

We have merged several tables and figures together and replaced complicated or confusing figures with tables. Additionally, we have moved several tables to the supplementary materials.

Follow the author guidelines on the formatting style of PLOSONE to the letter, including rules for formatting tables and figures as well as reference listing.

We look forward to receiving your revised manuscript.

Kind regards,

Nickson E. Otieno

Academic Editor

PLOS ONE

[RTI received financial support for this work from EPA’s Office of Research and Development (https://www.epa.gov/aboutepa/about-office-research-and-development-ord) under contract 68HERD20A0004/68HERH22F0010. The views presented are those of the authors and do not necessarily represent the views or policies of the US Environmental Protection Agency.].

We’ve added the following: “The funders participated in the study design, interpretation of results, and in manuscript preparation.”

3. We note that Figures 7 and 9 in your submission contain [map/satellite] images which may be copyrighted. All PLOS content is published under the Creative Commons Attribution License (CC BY 4.0), which means that the manuscript, images, and Supporting Information files will be freely available online, and any third party is permitted to access, download, copy, distribute, and use these materials in any way, even commercially, with proper attribution. For these reasons, we cannot publish previously copyrighted maps or satellite images created using proprietary data, such as Google software (Google Maps, Street View, and Earth). For more information, see our copyright guidelines: http://journals.plos.org/plosone/s/licenses-and-copyright.

Figures 7 and 9 are original work from this study and do not require copyright by anyone external.

1. You may seek permission from the original copyright holder of Figures 7 and 9 to publish the content specifically under the CC BY 4.0 license.

Additional Editor Comments (if provided):

Reviewers' comments:

Reviewer's Responses to Questions

Comments to the Author

1. Is the manuscript technically sound, and do the data support the conclusions?

Reviewer #1: Yes

Reviewer #2: No

2. Has the statistical analysis been performed appropriately and rigorously?

Reviewer #1: Yes

Reviewer #2: No

3. Have the authors made all data underlying the findings in their manuscript fully available?

Reviewer #1: Yes

Reviewer #2: No

4. Is the manuscript presented in an intelligible fashion and written in standard English?

Reviewer #1: Yes

Reviewer #2: Yes

5. Review Comments to the Author

Reviewer #1: Section Page # Para # Line # Comment/suggestion/question

Abstract 1 2 1 “……. fuel put pressure on ……..” instead of “….fuel influence the ……”

Suggested change made.

Introduction 3 3 2 Check that “frequency of data collection” and “time intervals of representation” are not used in the same context?

Fixed.

4 3 “…….reasons for these …….”

Suggested change made.

4 Table 1 For Global Cropland Expansion in the 21st Century (Potapov), the data is Landsat GLAD ARD an

---

## [Editor Report · Decision Letter 1]

4 Nov 2024

Tracking cropland transitions: a comparative analysis of U.S. land cover change data

PONE-D-24-19267R1

Dear Dr. Gray Martin,

We’re pleased to inform you that your manuscript has been judged scientifically suitable for publication and will be formally accepted for publication once it meets all outstanding technical requirements.

Kind regards,

Nickson E. Otieno

Academic Editor

PLOS ONE

Additional Editor Comments (optional):

Thank you for revising the manuscript and for adequately addressing most of the concerns and incorporating suggestions offered by the reviewers in improving your manuscript in content as well as structure.

In its improved form, the article may be acceptable if you are willing to make the following additional minor changes, point by point, to further improve it to standards suitable to PLOS ONE.

Given that the review comments and your responses as detailed below:

*Reviewer # 1*

*It must be adequately justified why the NRI and Potapov datasets were included in the article since they received much less treatment, or appear to be not as central to the comparisons and discussion as are the LCMAP and Lark et al components. Similarly, NAIP and NLCD datasets seem to be only of peripheral significance to the study.*

*Author response:*

*This is clarified in the revisions. The Lark vs. LCMAP comparison appears to be central because the Lark estimates (2020) have been driving much of the debate and the discussion in the literature. So we felt it was appropriate to focus on that comparison. Additionally, given the similarities in both the underlying datasets (CDL and LCMAP) and the processing methodologies, we were able to perform a pixelwise comparison between Lark and LCMAP that we could not reasonably do for NRI and Potapov as well. *

*The NRI is generally considered the best overall estimate, but it is not a remotely sensed dataset and thus is not available at the pixel scale that the Lark et al. (2020) and LCMAP datasets are. Potapov is included because it is the most recent of the four estimates and because it has a high spatial resolution*

*Reviewer #2*

*Overall the study seems to be more of a comparison between LCMAP and Lark datasets with NRI and Potapov datasets being forced into the comparison.*

*Author response:*

*Direct pixel-level comparison, as well as annual quantities of cropland change, between LCMAP and both NRI and Potapov are confounded by the temporal availability of these datasets. We have added an explanation of this in the methods.*

Would it not be wiser to leave the NRI narrative out altogether since, in your own admission, it has the lowest relevance in terms of both pixel-scale as well as centrality to the comparative nature of your study

Furthermore:

*Reviewer #2*

*: “We found that most of the pixel-level disagreements (86%) were due to definitional differences among datasets, whereas the remainder (14%) were from a variety of causes including spatial allocation.” this gives the impression that the comparison is between all the four datasets when that is clearly not the case.*

*Your response:*

*Clarified that the pixel-level disagreements only apply to the comparison between LCMAP and Lark et al. given that we did not perform a pixel-level comparison with the other two datasets.*

This might be a further reason for you to leave out the NRI narrative

*Reviewer #2:*

*Martin 11: Table 4 - Is there a reason “permanent managed pasture” is not included in cropland in the Potapov dataset when this subclass is included in cropland in LCMAP dataset? Or why “perennial woody crops including orchards and vineyards” is not included in cropland in the Potapov dataset when this subclass is included in cropland in Lark dataset? There’s mention of homogenisation in the discussion section Martin 27 paragraph 1 “To make a direct comparison we therefore harmonised—to the extent possible …” but there does not seem to be any homogenisation of the classes done. How would the results of the study change if the definitions are harmonised? Isn’t this what the study aims to do?*

*Your response:*

*The purpose of this study is not to perfectly harmonize these datasets. The dataset from Potapov et al. 2022 was made with a set of definitional assumptions that we cannot change after the fact. We note definitional differences where they occur as they impact the estimated cropland change, but we are unable to change the class definitions used to create the dataset. It is not possible—or at least, it is certainly outside the scope of this study—to rederive the Potapov et al. cropland extent maps with a definition of cropland that includes perennial woody crops or excludes managed hay. *

Perhaps it would help if you include this fact within the methodology or the discussion part of the manuscript so that your impression of harmonization puts your results into the context of your methodlogical reality, this is what the reviewer presumably had in mind, and was suggesting you do. I do realize of course that you have also now added a sentence about this in the last paragraph of the Introduction in your revised version, but make it a bit more explicit in methods or discussion (or both!).

In addition:

Remove all vertical lines from all tables, according to author guidelines)Make all number data in all table centered both within column and within each cellHave different font sizes to distinguish headings from subheadingsTable captions and figure legends must have sufficiently detailed information to stand alone,  ad without having any foot notesAvoid having any bold-face font in any table, whether column heading or row heading (just the title only
---

## [Editor Report · Acceptance letter]

PONE-D-24-19267R1

PLOS ONE

Dear Dr. Martin,

I'm pleased to inform you that your manuscript has been deemed suitable for publication in PLOS ONE. Congratulations! Your manuscript is now being handed over to our production team.

Kind regards,

on behalf of

Dr. Nickson E. Otieno

Academic Editor

PLOS ONE